# *Aspalathus linearis* suppresses cell survival and proliferation of enzalutamide-resistant prostate cancer cells via inhibition of c-Myc and stability of androgen receptor

Bi-Juan Wang[1☯], Shih-Han Huang[1,2☯], Cheng-Li Kao[3,4☯], Christo J. F. Muller[5,6,7], Ya-Pei Wang[1], Kai-Hsiung Chang[1], Hui-Chin Wen[1], Chien-Chih Yeh[8,9], Li-Jane Shih[8,10], Yung-Hsi Kao[2], Shu-Pin Huang[11,12,13,17], Chia-Yang Li[14], Chih-Pin Chuu[1,2,15,16]*

1 Institute of Cellular and System Medicine, National Health Research Institutes, Miaoli County, Taiwan, 2 Department of Life Sciences, National Central University, Taoyuan City, Taiwan, 3 Division of Urology, Departments of Surgery, Tri-Service General Hospital, National Defense Medical Center, Taipei, Taiwan, 4 Division of Urology, Department of Surgery, Taoyuan Armed Forces General Hospital, Taoyuan, Taiwan, 5 Biomedical Research and Innovation Platform (BRIP), South African Medical Research Council, Tygerberg, South Africa, 6 Division of Medical Physiology, Faculty of Health Sciences, Stellenbosch University, Tygerberg, South Africa, 7 Department of Biochemistry and Microbiology, University of Zululand, KwaDlangezwa, South Africa, 8 Department of Education and Medical Research, Taoyuan Armed Forces General Hospital, Taoyuan, Taiwan, 9 Division of Colon and Rectal Surgery, Department of Surgery, Tri-Service General Hospital, National Defense Medical Center, Taipei, Taiwan, 10 Graduate Institute of Medical Science, National Defense Medical Center, Taipei, Taiwan, 11 Department of Urology, School of Medicine, College of Medicine, Kaohsiung Medical University, Kaohsiung Taiwan, 12 Department of Urology, Kaohsiung Medical University Hospital, Kaohsiung Medical University, Kaohsiung, Taiwan, 13 Graduate Institute of Clinical Medicine, College of Medicine, Kaohsiung Medical University, Kaohsiung, Taiwan, 14 Graduate Institute of Medicine, College of Medicine, Kaohsiung Medical University, Kaohsiung, Taiwan, 15 PhD Program for Aging and Graduate Institute of Basic Medical Science, China Medical University, Taichung City, Taiwan, 16 Biotechnology Center, National Chung Hsing University, Taichung City, Taiwan, 17 Institute of Biomedical Science, National Sun Yat-sen University, Kaohsiung City, Taiwan

☯ These authors contributed equally to this work.

* cpchuu@nhri.org.tw

**Data Availability Statement:** All relevant data are within the paper and its Supporting Information files.

## Abstract

Enzalutamide, a nonsteroidal antiandrogen, significantly prolonged the survival of patients with metastatic castration-resistant prostate cancer (CRPC). However, patients receiving enzalutamide frequently develop drug resistance. Rooibos (*Aspalathus linearis*) is a shrub-like leguminous fynbos plant endemic to the Cedarberg Mountains area in South Africa. We evaluated the possibility of using a pharmaceutical-grade green rooibos extract (GRT, containing 12.78% aspalathin) to suppress the proliferation and survival of enzalutamide-resistant prostate cancer (PCa) cells. Treatment with GRT dose-dependently suppressed the proliferation, survival, and colony formation of enzalutamide-resistant C4-2 MDV3100r cells and PC-3 cells. Non-cancerous human cells were more resistant to GRT treatment. GRT suppressed the expression of proteins involved in phosphoinositide 3-kinase (PI3K)-Akt signaling, androgen receptor (AR), phospho-AR (Ser81), cyclin-dependent kinase 1 (Cdk1), c-Myc and Bcl-2 but increased the expression of apoptotic proteins. Overexpression of c-Myc antagonized the suppressive effects of GRT, while knockdown of c-Myc increased the sensitivity of PCa cells to GRT treatment. Expression level of c-Myc correlated to resistance of

**Funding:** This research was funded by Ministry of Science and Technology in Taiwan (MOST 109-2923-B-400-002-MY3), intramural grant from National Health Research Institutes (CS-110/111-PP-03), collaboration grant between National Health Research Institutes and Kaohsiung Medical University (NHRIKMU-111-I002), and the National Research Foundation (NRF) of South Africa (IRG-Taiwan/South African Research Cooperation Programme; Grant Number 98854). The funders had no role in study design, data collection and analysis, decision to publish, or preparation of the manuscript.

**Competing interests:** The authors have declared that no competing interests exist.

**Abbreviations:** ADT, Androgen deprivation therapy; AR, androgen receptor; Cdk1, cyclin-dependent kinase 1; CRPC, castration-resistant prostate cancer; DHEA, dyhydroepiandrosterone; GRT, green rooibos extract; HPLC, high performance liquid chromatography; PCa, prostate cancer; PSA, prostate specific antigen; PI3K, phosphoinositide 3-kinase; PTEN, phosphatase and tensin homolog; SKP2, S phase kinase-associated protein 2.

PCa cells to GRT treatment. Additionally, immunofluorescence microscopy demonstrated that GRT reduced the abundance of AR proteins both in nucleus and cytoplasm. Treatment with cycloheximide revealed that GRT reduced the stability of AR. GRT suppressed protein expression of AR and AR's downstream target prostate specific antigen (PSA) in C4-2 MDV3100r cells. Interestingly, we observed that AR proteins accumulate in nucleus and PSA expression is activated in the AR-positive enzalutamide-resistant PCa cells even in the absence of androgen. Our results suggested that GRT treatment suppressed the cell proliferation and survival of enzalutamide-resistant PCa cells via inhibition of c-Myc, induction of apoptosis, as well as the suppression of expression, signaling and stability of AR. GRT is a potential adjuvant therapeutic agent for enzalutamide-resistant PCa.

## Introduction

Prostate cancer (PCa) is the second most frequently diagnosed cancer of men and the fifth most common cancer overall in the world. Nearly 1,200,000 new cases were diagnosed every year in the world. More than 80% of patients died from PCa developed bone metastases. Compared to localized PCa, metastatic PCa has a poorer prognosis and higher mortality rate, with a 5-year survival less than 30%. Androgen deprivation therapy (ADT) is the standard treatment for metastatic prostate cancer (PCa). However, a majority of patients receiving ADT eventually develop castration-resistant prostate cancer (CRPC) within 1–3 years. Enzalutamide (formerly called MDV3100) is a nonsteroidal antiandrogen introduced in 2012 for treatment of metastatic CRPC. Enzalutamide significantly prolonged the survival of patients with metastatic CRPC. The enzalutamide treatment group has a longer median overall survival (18.4 vs. 13.6 months), a greater decrease in prostate-specific antigen (PSA) level by 50% or more (54% vs. 2%), and a longer time to radiographic progression-free survival (8.3 vs. 2.9 months) as compared to the placebo group [1]. However, most patients receiving enzalutamide treatment develop drug resistance after a median of 18 months [2]. New therapeutic agents are therefore needed.

Rooibos (*Aspalathus linearis*) is a shrub-like leguminous fynbos plant endemic to the Cedarberg Mountains area in the Western Cape Province of South Africa. Rooibos is becoming more and more popular and is consumed as daily beverage world-wide [3, 4]. The traditional rooibos tea product is produced by fermentation (oxidation) of the chopped grinded plant material with the leaf color changing from green to red-brown. Recently, more and more attentions are paid to preserve the antioxidant ingredient in rooibos. The "unfermented" or so-called green rooibos is produced by limiting oxidation of polyphenols during processing [3, 4]. Rooibos tea does not contain caffeine but it contains various types of polyphenols. Major compounds of rooibos include dihydrochalcone, aspalathin, orientin and isoorientin [5]. These compounds are *C*-glucosides which are poorly absorbed but exhibit strong anti-oxidative activity [6]. We previously reported that treatment with a pharmaceutical-grade green rooibos extract (GRT, containing 12.78% aspalathin) suppresses Akt signaling and therefore inhibits cell proliferation, cell survival, and tumor growth in CRPC cell line LNCaP-104-R1 [7]. Additionally, we observed that GRT treatment suppressed the migration and invasion of CRPC cells via inhibition of YAP signaling and paxillin [8]. PI3K/AKT signaling has been reported to play an essential role in the proliferation and survival of enzalutamide-resistant PCa cells [9], we therefore investigated if treatment with GRT can inhibit the proliferation and survival of enzalutamide-resistant PCa cells.

## Materials and methods

### Rooibos extract GRT

A pharmaceutical-grade aspalathin-rich green rooibos extract (GRT$^{TM}$) was used in all experimental work from the same batch as previously described [7]. The phenolic component has previously been confirmed by high performance liquid chromatography (HPLC) [7] (S1 Fig). The major flavonoids of the extract (comprised 20.63% of the extract) include aspalathin (12.78%), nothofagin (1.97%), isoorientin (1.47%), orientin (1.26%) and bioquercetin (quercetin-3-O-robinobioside; 1.04%). Compounds which are less than 1% include luteoloside, vitexin, isovitexin rutin, hypersoide, isoquercitrin and the phenolic precursor, Z-2-(β-D-glucopyranosyloxy)-3-phenylpropenoic acid. The pinitol and glucose content of the extract were 1.92 and 0.50%, respectively (quantification by GC-MS by Central Analytical Facility, Stellenbosch University, Stellenbosch, South Africa).

### Cell culture

LNCaP C4-2 MDV3100r cells were derived from parental androgen-independent LNCaP C4-2 cells (ATCC CRL-3314). LNCaP C4-2 and LNCaP C4-2 MDV3100r cells were maintained in HyClone RPMI-1640 phenol red free medium (Cytiva, Marlborough, MA, USA) supplemented with 10% hormone deprivation FBS (Corning, Corning, NY, USA), penicillin (100 U/ml) and streptomycin (100 μg/ml) (Merck Millipore, Burlington, MA, USA). Medium for C4-2 MDV3100r contained 10 μM enzalutamide plus 20 nM DHEA (dehydroepiandrosterone). PC-3 cells were maintained in HyClone DMEM medium with 10% FBS, penicillin (100 U/ml) and streptomycin (100 μg/ml).

### Cell proliferation assay and chemicals

LNCaP C4-2, LNCaP C4-2 MDV3100r, PC-3 cells were seeded at a density of $5 \times 10^3$ cells/well in 96-well plates with 100 μl culture medium containing 10% FBS and increasing concentration (0, 10, 25, 50, 75, 100 μg/ml) of GRT for 96 h. Relative cell number was analyzed by measuring the DNA content of cell lysates with Hoechst dye 33258-based 96-well proliferation assay (Sigma, St. Louis, MO, USA) as described previously [10]. All readouts were normalized to the average of the control condition in each individual experiment. The experiment was repeated at least three times. Eight wells were used for each condition.

### Immunofluorescence staining

Immunofluorescence staining analysis has been described previously [11]. Briefly, LNCaP C4-2 MDV3100r cells were seeded on chamber slide overnight. The cells were treated with GRT for 96 h, washed by cold PBS, then fixed in 4% paraformaldehyde for 15 minutes at room temperature, followed by cold PBS wash for 5 minutes three times. Cells were incubated in blocking buffer (5% BSA, 0.1% triton X-100 in PBS) for an hour at room temperature, then washed by PBS for 5 minutes three times. Cells were incubated with primary antibodies overnight at 4˚C. After cold PBS wash for 5 minutes three times, cells were incubated with secondary antibody (FITC) for an hour at room temperature. Following several washes, cells were stained using 4′,6-diamidino-2-phenylindole (DAPI) (5 mg/ml) for 15 minutes at room temperature. After staining, cells were washed, mounted and sealed. Images of the cells were captured at a magnification of 1000× fluorescent microscope with fluorescence secondary antibodies: goat anti mouse 488 (green) and goat anti rabbit 594 (red) (Sigma).

## Soft agar colony formation assay

A total of 8,000 LNCaP C4-2 MDV3100r cells were suspended in 0.3% low melting agarose (Lonza) with 10% FBS in DMEM medium and then layered on top of 3 ml of 0.5% low melting agarose plus 10% FBS in DMEM medium in 6 cm dishes. Cells were allowed to grow at 37˚C with 5% CO2 for 14 days. The plates were stained with 0.005% crystal violet in 30% ethanol for 6 h.

## Western blot analysis

For determining the effect of GRT treatment on protein expression in C4-2 MDV3100r cells, cells were lysed in MCLB (Mammalian Cell Lysis Buffer) containing 50 mM Tris-HCl pH 8.0, 0.5% NP-40, 150 mM NaCl, 5 mM EDTA with Na3VO4, DTT, protease inhibitors and phosphatase inhibitors cocktail. The signal detection was performed using chemiluminescence (ECL) and Prime Western Blotting detection reagent (Fisher Scientific, Pittsburgh, PA, USA). Antibodies against caspase 3, Akt, phospho-Akt (T308), phospho-Akt (S473), CDK1, PDK1, phospho-PDK1 (S241), cyclin D1, PARP were purchased from Cell Signaling (Dancers, MA, USA). Antibodies detecting PI3K α, PI3K β, PI3K γ were purchased from Millipore (Burlington, MA, USA), while antibody against Skp2 was from Santa Cruz (Dallas, TX, USA). Antibody detecting BCL-2 was purchased from BD (Franklin Lakes, NJ, USA), while Novus (Littleton, CO, USA) supplied antibody detecting GAPDH and β-actin. Antibodies against AR and Myc were purchased from Abcam (Cambridge, MA, U.S.A), while antibody detecting phospho-AR (S81) was from Millipore. Relative expression level of proteins was quantified by ImageQuant LAS4000 (GE Healthcare-Biosciences, Pittsburgh, PA).

## Comet assay

LNCaP C4-2 MDV3100r cells were seeded overnight in a 6 cm dish with a density of $1 \times 10^5$ cells per well in 5 ml of culture medium. Cells were treated with GRT (dissolved in 60% ethanol) at increasing concentrations (0, 50, 75, 100 μg/mL) for 48 h on the 2nd day. Cells were collected and re-suspended with ice-cold PBS without Mg2+ and Ca2+, followed by mixing with agarose (1:10) and transferred immediately to Comet Slides following the manufacture's instruction. Comet Slides were kept at 4˚C for 15 minutes protected from the light and were carefully transferred to ice-cold lysis buffer (Trevigen, Gaithersburg, MD, USA) at 4˚C for 30–60 minutes in the dark. Lysis buffer was then discarded and re-filled with ice-cold alkaline solution at 4˚C for 30 min protected from light. The alkaline solution was then decanted from the container and replaced with ice-cold TBE buffer and Comet Slides were immersed twice for 5 minutes. Comet Slides were carefully transferred to an electrophoresis chamber with ice-cold TBE buffer to approximately cover the Comet Slides, and then 35 V applied for 15 minutes. Comet Slides were carefully transferred from the chamber to ice-cold deionized water and immersed three times for 2 minutes. Ice-cold deionized water was then aspirated and replaced with ice-cold 70% ethanol for 5 minutes. After drying, 100 μl Vista Green DNA Dye was added to the agarose on the slide and the images were recorded by fluorescence microscopy.

## Knockdown c-Myc with siRNA and c-Myc overexpression

Human c-Myc siRNA (Human c-Myc ON-TARGET plus SMART pool) and randomly scrambled sequence control were purchased from Dharmacon (Lafayette, CO, USA). The transfection procedure was performed using lipofectamine RNAiMAX (Invitrogen, Carlsbad, CA, U. S.A.) according to the manufacturer's recommended protocol. Cells were seeded at a density

of $3 \times 10^5$ cells/well in 2.5ml complete medium in 6-well plates overnight. 40 nM siRNA were used for scramble or c-Myc knockdown. For overexpression of c-Myc, PC-3 cells were transfected with c-Myc plasmid, a generous gift from Shutsung Liao's lab at the University of Chicago as previously described [12]. The PLNCX2 plasmid was used as controls for c-Myc overexpression. PolyJet™ *in vitro* DNA transfection reagent was used for the transfection (SL100688, SigmaGen Laboratories, Rockville, MD, USA). Western blotting was used to confirm the knockdown or overexpression of c-Myc.

## Statistical analysis

Data are represented as the mean ± SD from three independent experiments. Student's T-test was performed to test statistical significance. $P < 0.05$ was considered statistically significant.

## Results

### GRT treatment suppressed the proliferation of enzalutamide-resistant C4-2 MDV3100r and PC-3 PCa cells

We establish an enzalutamide (MDV3100)-resistant prostate cancer (PCa) cell line derived from LNCaP C4-2 by culturing the C4-2 cells in medium containing 10 μM enzalutamide and 20 nM dyhydroepiandrosterone (DHEA) for more than 3 months. The derived cell line, LNCaP C4-2 MDV3100r, was much more resistant to the enzalutamide treatment as compared to parental cell line (Fig 1A and 1B). The C4-2 MDV3100r cells were maintained in medium containing 10 μM enzalutamide for all the following experiments. Another commonly-used androgen receptor (AR)-negative PCa cell line, PC-3, was also very resistant to enzalutamide treatment (Fig 1C). The morphology of C4-2 MDV 3100r cells was slightly different from that of C4-2 cells (Fig 1D and 1E). C4-2 MDV 3100r cells were not as elongated as C4-2.

We next examined if treatment with GRT can suppress the proliferation of C4-2 MDV3100r and PC-3 cells. Treatment with GRT dose-dependently reduced the proliferation of C4-2 MDV3100r cells (Fig 2A) and PC-3 cells (Fig 2B). The $IC_{50}$ for GRT to suppress the proliferation of C4-2 MDV3100r and PC-3 cells was 98.5 and 226.6 (μg/ml), respectively. GRT treatment altered the morphology of C4-2 MDV3100r cells, C4-2 3100r cells became round-shape under GRT treatment (Fig 2C). For comparison, we examined if GRT exhibit suppressive effects on non-cancerous cells. We treated HUVEC (human umbilical vein endothelial cells) and HEK293 (human embryonic kidney cells 293) with increasing concentration of GRT (Fig 2D). Proliferation of HUVEC cells was not affected by 10–75 μg/ml GRT and was relatively resistant to 100 μg/ml GRT, while HEK293 was not affected by GRT at all. This observation suggested that enzalutamide-resistant PCa cells were more sensitive to GRT treatment than non-cancerous human cell lines.

### GRT treatment induced apoptosis in enzalutamide-resistant C4-2 MDV3100r PCa cells

To confirm the anti-cancer activity of GRT on enzalutamide-resistant PCa cells, we performed soft-agar colony formation assay. Treatment with GRT (0, 50, 75, 100 μg/ml) dose-dependently reduced the number and size of colonies of C4-2 MDV3100r cells in the soft agar plates (Fig 3A). We next investigated if GRT induced apoptosis in C4-2 MDV3100r cells. Comet assay revealed that treatment with GRT dose-dependently increased the DNA strand break in C4-2 MDV3100r cells (Fig 3B and 3C), indicating that GRT induced apoptosis in C4-2 MDV3100r cells.

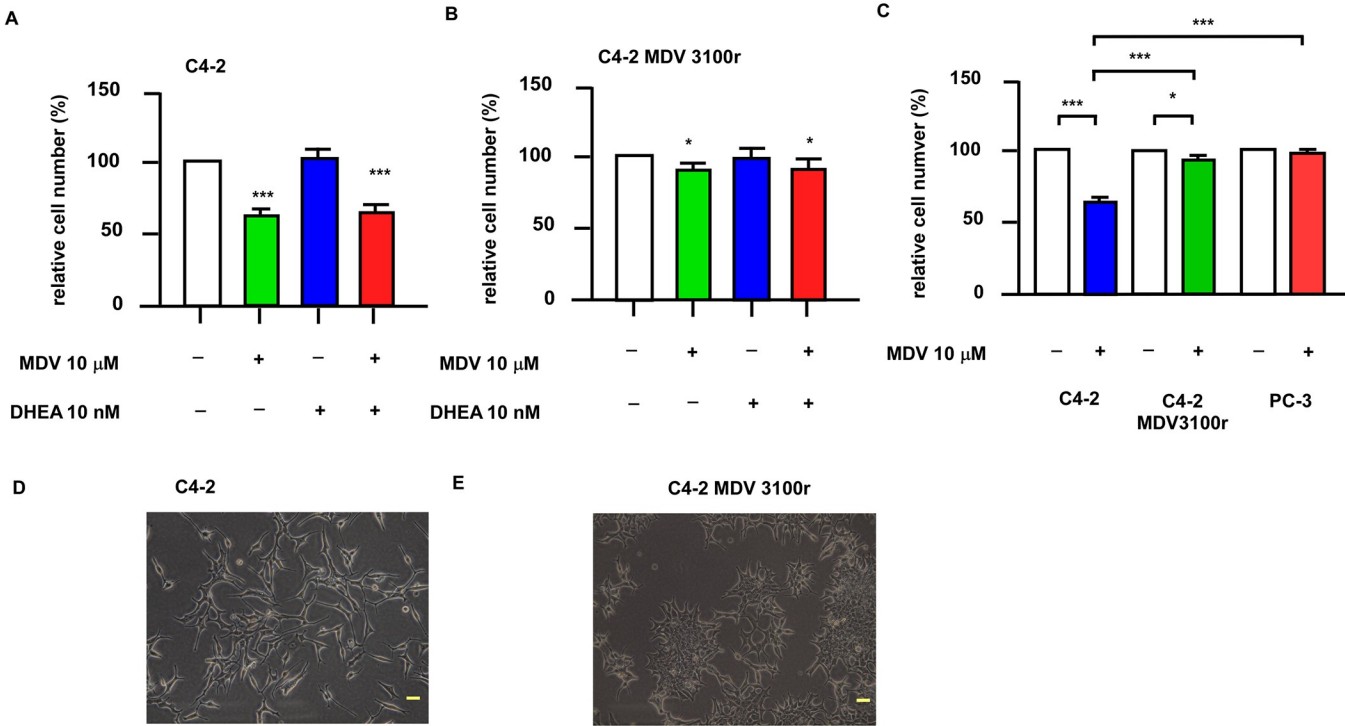

**Fig 1. Characterizing the response of enzalutamide-resistant C4-2 MDV3100r and PC-3 cell lines to treatment of GRT and DHEA.** The enzalutamide-resistant C4-2 MDV3100r cell lines was derived from parental LNCaP C4-2 by culturing the cell in medium containing 10 μM enzalutamide and 20 nM DHEA for at least 3 months. The effect of enzalutamide (MDV-3100) treatment in the presence or absence of DHEA on the proliferation of C4-2 (A) and C4-2 MDV3100r (B) for 96 h was examined by Hoechst 33258 proliferation assay. (C) The proliferation of C4-2, C4-2 MDV3100r, and PC-3 under 0 or 10 μM MDV for 96 h was examined by proliferation assay. The morphology of C4-2 and C4-2 MDV3100r were show in (D) and (E) under light microscope. The magnification of the microscope is 1,000X and the yellow scale bar represented 100 μm.

## GRT treatment suppressed c-Myc, AR, PI3K-AKT signaling proteins and cell cycle regulatory proteins

To determine how GRT repressed the proliferation and survival of enzalutamide-resistant cells, we studied the effects of GRT on expression of signaling proteins involved in regulation of cell cycle, apoptosis, cell survival and androgen receptor (AR) signaling (Fig 4). GRT treatment dose-dependently suppressed the protein expression level of Akt, PI3K p110α, PI3K p110γ, phosphoinositide-dependent kinase 1 (PDK1), phospho-PDK1 (S241), S phase kinase-associated protein 2 (Skp2), cyclin D1, AR, phospho-AR (S81), cyclin-dependent kinase 1 (Cdk1), pro-PARP, c-Myc and Bcl-2 but increased protein abundance of caspase 3 and cleaved-PARP in C4-2 MDV3100r cells (Fig 4).

## GRT suppressed proliferation of enzalutamide-resistant PCa cells via inhibition of c-Myc

The c-Myc protein is essential in regulating tumor growth, metastasis, disease progression, and metabolism of PCa cells. Since GRT treatment significantly repressed the protein expression of c-Myc, we explored if c-Myc is one of the main targets of GRT. We overexpressed c-Myc in C4-2 MDV3100r cells (Fig 5A). C4-2 MDV3100r cells with c-Myc overexpression were more resistant to GRT treatment (Fig 5B). The $IC_{50}$ for GRT to suppress the proliferation of C4-2 MDV3100r cells with control vector and C4-2 MDV3100r overexpressing c-Myc cells was 105.3 and 141.1 (μg/ml), respectively. On the other hand,

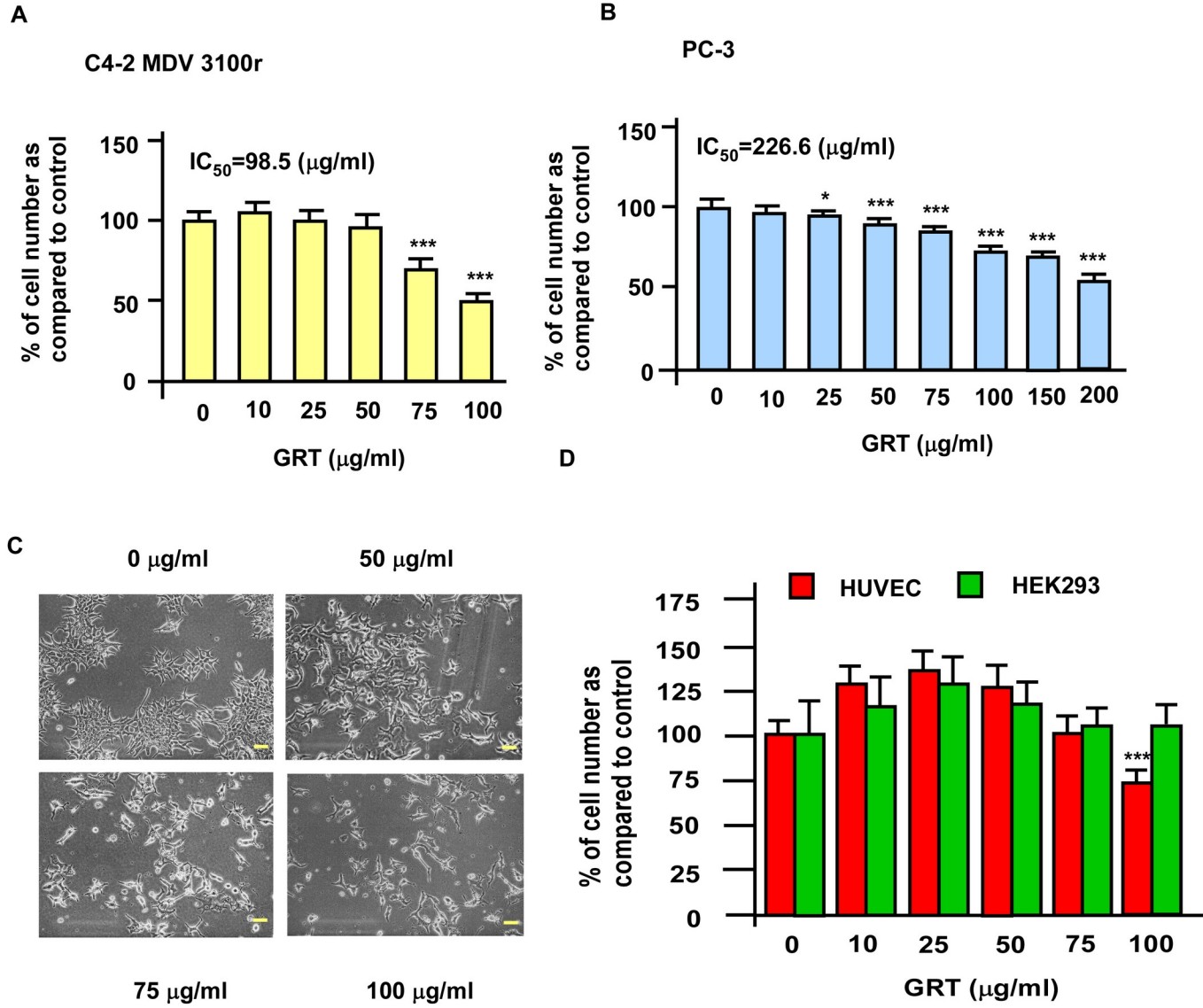

**Fig 2. Effects of GRT on proliferation of enzalutamide-resistant C4-2 MDV3100r, PC-3, and non-cancerous human cell lines.** Proliferation of LNCaP C4-2 MDV3100r (A) and PC-3 (B) being treated with increasing concentration of GRT (0–100 μg/ml for C4-2 MDV3100r cells and 0–200 μg/ml for PC-3 cells) for 96 h was determined by Hoechst 33258 proliferation assay. (C) Morphology of C4-2 MDV3100r cells under treatment of different concentrations of GRT (0, 50, 75, 100 μg/ml) was examined by microscope. The magnification of the microscope is 100X and the yellow scale bar represented 100 μm. (D) Proliferation of HUVEC and HEK293 cells being treated with increasing concentration of GRT (0–100 μg/ml) was determined by Hoechst 33258 proliferation assay. Asterisk $^*$, $^{**}$ and $^{***}$ represents statistically significant difference $p < 0.05$, $p < 0.01$ and $p < 0.001$, respectively, between the two groups of cells being compared.

knockdown of c-Myc with siRNA in PC-3 cells increased the sensitivity of PC-3 cells to GRT treatment (Fig 6A and 6B). We compared the c-Myc protein level in C4-2 MDV3100r and PC-3. PC-3, which was more resistant to GRT treatment, expressed higher level of c-Myc protein as compared to C4-2 MDV3100r (Fig 6C). GRT treatment caused a more dramatic reduction of c-Myc in C4-2 MDV3100r cells as compared to PC-3 cells (Fig 6C). These observations suggested that c-Myc is one of the main targets of GRT in enzalutamide-resistant PCa cells.

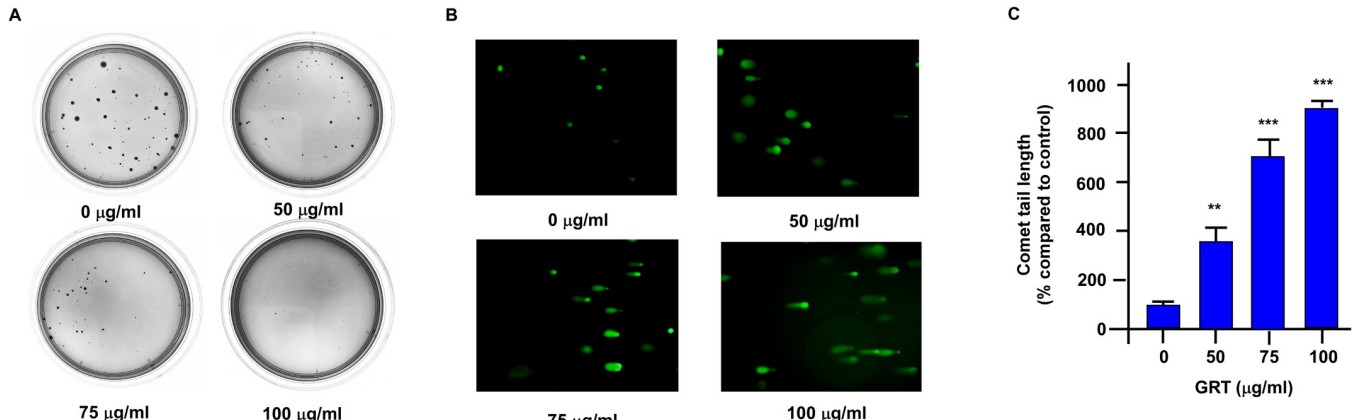

**Fig 3. GRT treatment reduced soft agar colony formation of C4-2 MDV3100r cells and induced apoptosis.** (A) Anticancer activity of GRT on C4-2 MDV3100r cells was confirmed by measuring the formation of colonies on soft agar for C4-2 MDV3100r cells being treated with GRT (0, 50, 75 and 100 μg/ mL, being added into the medium). The culture medium was changed twice a week for four weeks. (B) DNA damage caused by GRT treatment in C4-2 MDV3100r cells was determined using the Comet assay. Green fluorescent indicated broken DNA fragments in C4-2 MDV3100r cells being treated with increasing concentration of GRT (0, 50, 75, 100 μg/ml). (C) Tail extent moment of C4-2 MDV3100r cells in Comet assay was quantified. Data wa presented as mean ± SD in triplicate experiments. Asterisk ** and *** represents statistically significant difference p < 0.01 and p < 0.001, respectively, between the two groups of cells being compared.

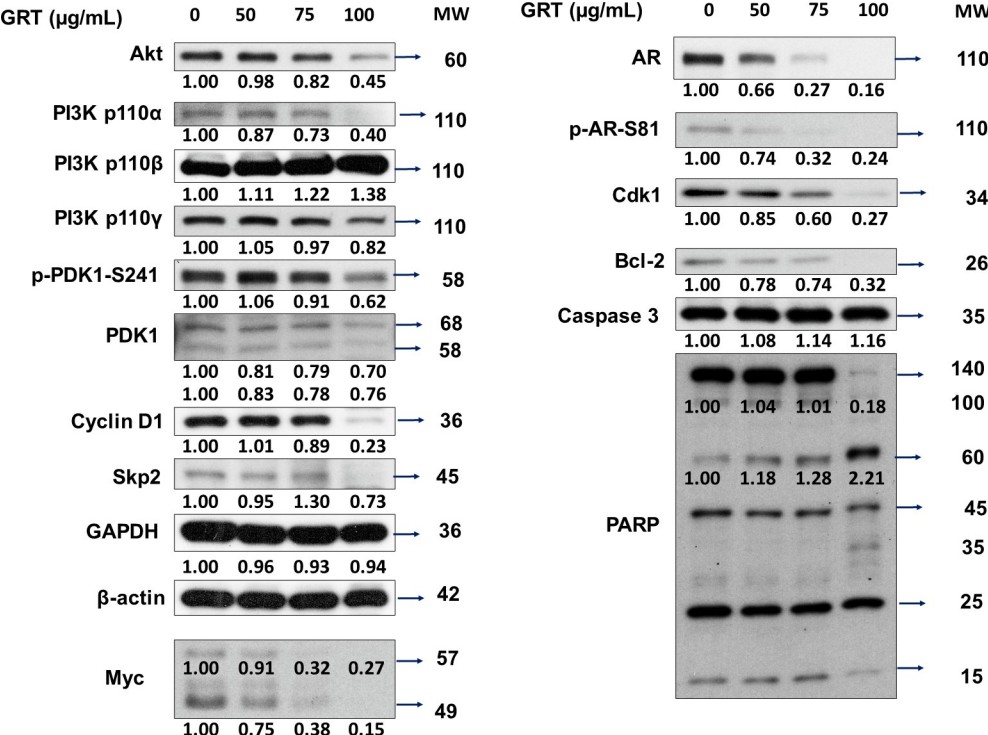

**Fig 4. Western blotting analysis of signaling proteins affected by GRT treatment in C4-2 MDV3100r cells.** The C4-2 MDV3100r cells were treated with increasing concentrations of GRT (0, 50, 75, and 100 μg/mL) for 48 h. Expression of Akt, PI3K p110α, PI3K p110β, PI3K p110γ, PDK1, phospho-PDK1 (S241), Skp2, cyclin D1, AR, phospho-AR (S81), Cdk1, Myc, Bcl-2, caspase 3 and PARP proteins were determined by Western blotting. Expression level of GAPDH and β-Actin were used as loading control.

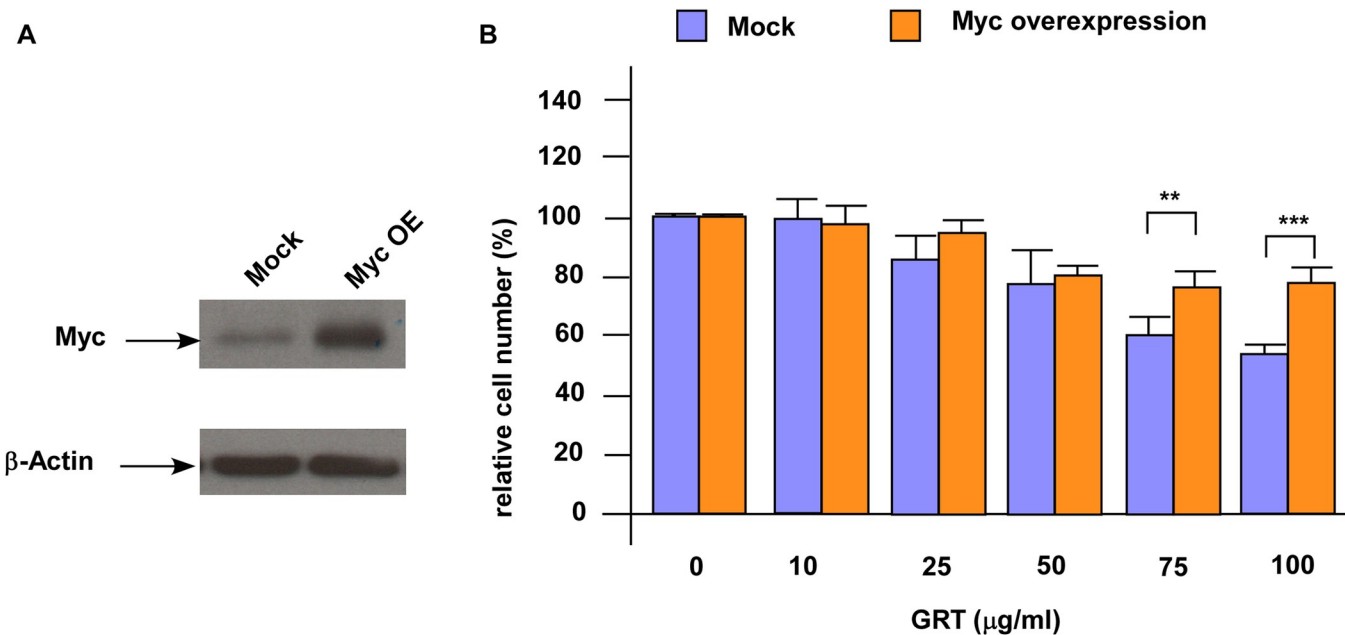

**Fig 5. Overexpression of c-Myc enhanced resistance of C4-2 MDV3100r cells against GRT treatment.** (A) Overexpression of c-Myc in LNCaP C4-2 MDV3100r cells was confirmed by Western blotting. (B) LNCaP C4-2 3100r cells with mock vector or c-Myc overexpression were treated with increasing concentration of GRT (0, 10, 25, 50, 75, 100 μg/ml) and cell proliferation was examined by proliferation assay. Asterisk ** and *** represents statistically significant difference $p < 0.01$ and $p < 0.001$, respectively, between the group as compared to control group or the denoted two groups being compared.

## GRT suppressed the stability, expression, and downstream signaling of AR in C4-2 MDV3100r cells

As we observed that GRT repressed the protein expression of AR and phospho-AR (S81) and AR plays an important role during the PCa progression, we used immunofluorescence microscopy to examine the effects of GRT on the distribution and abundance of AR protein in C4-2 MDV3100r cells. We did not examine PC-3 because PC-3 cells were AR-negative. The distribution of AR in LNCaP C4-2 MDV3100r cells was mainly located in the nucleus (Fig 7A). GRT treatment dose-dependently reduced the abundance of AR proteins in C4-2 MDV3100r cells (Fig 6A) as well as expression of AR mRNA (Fig 7B) in C4-2 MDV3100r cells. To determine if treatment with GRT reduces the stability of AR, we treated LNCaP C4-2 MDV3100r cells with cycloheximide in the presence or absence of GRT. GRT treatment accelerated the degradation of AR (Fig 7C). Additionally, GRT treatment repressed the protein expression of AR and AR's target gene prostate specific antigen (PSA) both in the absence or presence of androgen (Fig 7D). Interesting, we observed that AR proteins accumulated in nucleus and PSA expression was abundant in the enzalutamide-resistant PCa cells even in the absence of androgen (Fig 7D). The androgen-independent activation of AR and PSA may be important for the proliferation and survival of enzalutamide-resistant PCa cells.

## Discussion

Prostate cancer (PCa) patients receiving androgen ablation therapy usually develop castration-resistant prostate cancer (CRPC) within 2–3 years. Enzalutamide, a nonsteroidal antiandrogen, is an USA FDA-approved drug for treatment of metastatic CRPC. However, a great portion of PCa patients who initially respond to enzalutamide treatment develop drug resistance after a few months [13]. Mechanisms of enzalutamide resistance include AR amplification and

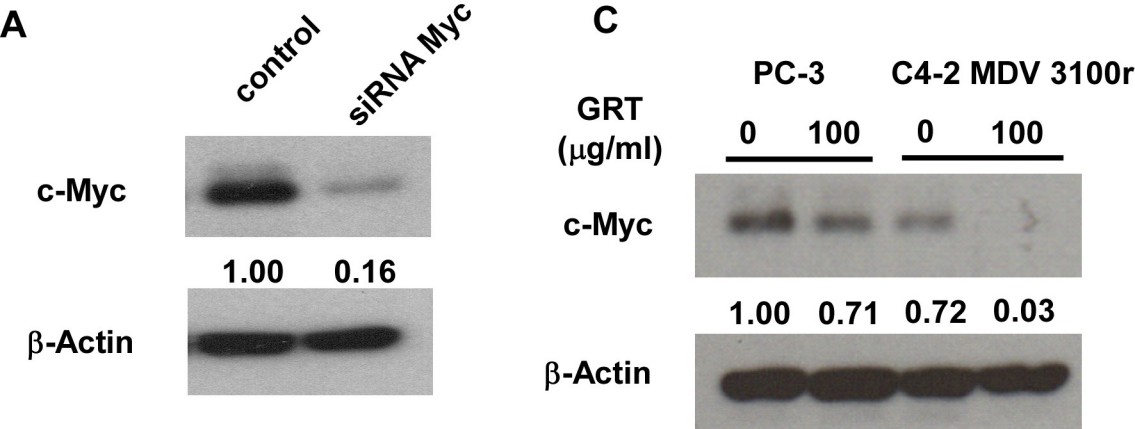

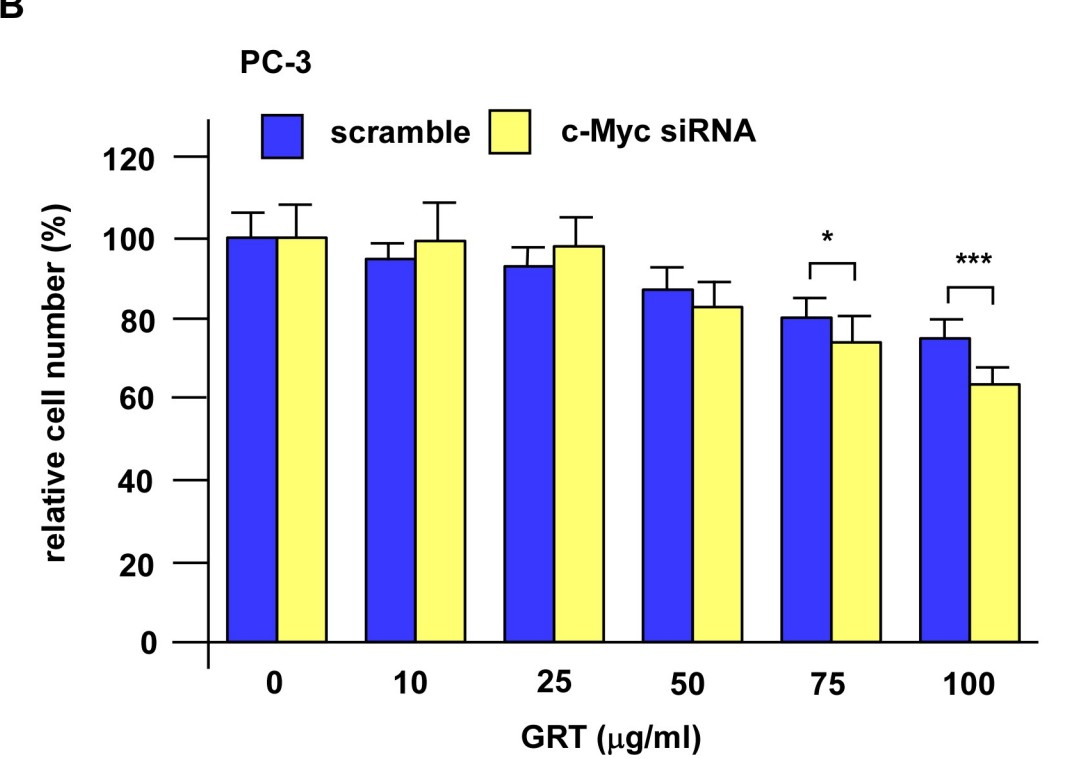

**Fig 6. Knockdown of c-Myc increased sensitivity of PC-3 cells to GRT treatment.** (A) Knockdown of c-Myc in PC-3 cells was confirmed by Western blotting. (B) PC-3 cells with control scramble or c-Myc knockdown were treated with increasing concentration of GRT (0, 10, 25, 50, 75, 100 μg/ml) and cell proliferation was examined by proliferation assay. Asterisk * and *** represents statistically significant difference $p < 0.05$ and $p < 0.001$, respectively, between the group as compared to control group or the denoted two groups being compared. (C) The protein expression of c-Myc in C4-2 MDV3100r or PC-3 cells being treated with or without 100 μg/ml was examined by Western blotting. The β-Actin was used as loading control.

overexpression, AR mutation, altered steroidogenesis, AR splice variants, autophagy mediated resistance, and activation of Wnt signaling [14]. New therapy for enzalutamide-resistant PCa is undoubtedly needed. In this study, we examined the suppressive effects of GRT, an aspalathin-rich green rooibos extract containing 12.78% aspalathin as major flavonoid, on

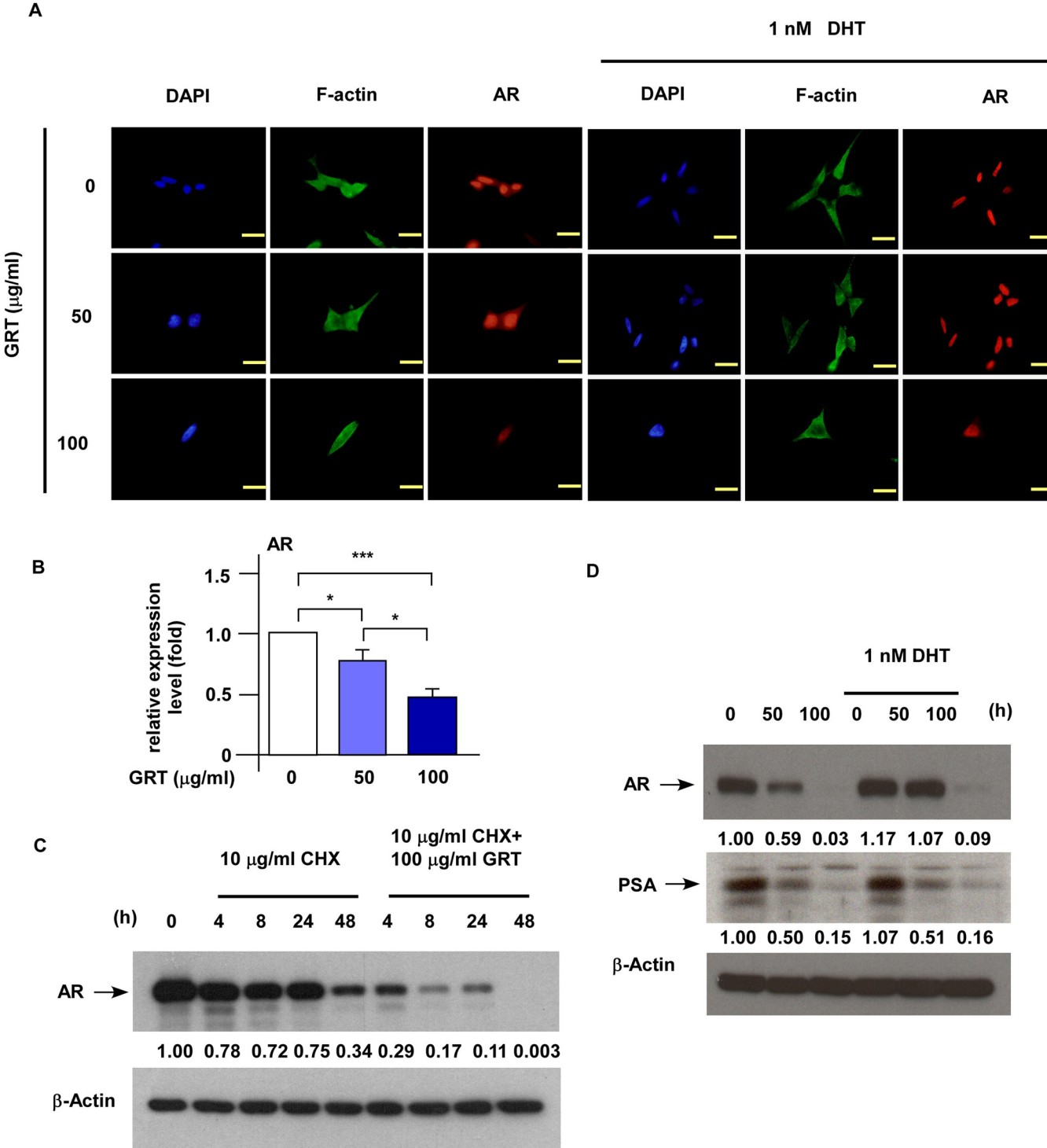

**Fig 7. GRT treatment suppresses the protein expression, stability, and downstream signaling of AR in C4-2 MDV3100r cells.** (A) C4-2 MDV3100r cells being treated with or without 1 nM DHT or increasing concentration of GRT (0, 50, 100 μg/ml) for 48 h was observed by immunofluorescent microscopy. The column from left to right shows DAPI (blue), F-actin (green), and AR (androgen receptor) (red) images. The magnification of the microscope is 100X and the yellow scale bar represented 100 μm. (B) Expression of AR mRNA in C4-2 MDV3100r cells being treated with 0, 50, 100 μg/ml was examined by qRT-PCR. Asterisk * and *** represents statistically significant difference $p < 0.05$ and $p < 0.001$, respectively, between the two groups of cells being compared. (C) C4-2 MDV3100r cells were treated with 10 μg/ml cycloheximide with or without 100 μg/ml GRT for 0, 4, 8, 24, 48 h. Expression of AR protein was examined by Western blotting and the β-actin was used as loading control. (D) LNCaP C4-2 MDV3100r cells were treated with or without 1 nM DHT along with increasing concentration of GRT (0, 50, 100 μg/ml) for 96 h. Abundance of AR and PSA protein was determined by Western blotting. The β-actin was used as loading control.

enzalutamide-resistant C4-2 MDV3100r and PC-3 cells. We observed that GRT effectively suppressed the proliferation, survival, and soft agar colony formation of enzalutamide-resistant C4-2 MDV3100r cells and PC-3 cells. The GRT treatment caused apoptosis in C4-2 MDV3100r cells within 48 h.

Phosphatase and tensin homolog (PTEN) is a negative regulator for phosphoinositide 3-kinase (PI3K)-Akt signaling pathway [15]. Deletion of PTEN was observed in 40–70% of PCa patients, resulting in upregulation of PI3K-Akt signaling. PI3K-Akt signaling plays an important role in the survival of PCa cells [16]. Phosphorylation of Akt is activated by PDK1 and mTOR kinase. We observed that GRT treatment dose-dependently suppressed the expression level of proteins involved in Akt signaling pathway, including Akt, PI3K α, PI3K γ, PDK1, phospho-PDK1 (S241) in C4-2 MDV3100r cells.

The oncoprotein c-Myc is commonly overexpressed in PCa [17]. Expression of c-Myc protein is elevated in CRPC [17] and promotes androgen-independent proliferation of CRPC cells [18]. Overexpression of c-Myc in PCa patients predicts biochemical recurrence of tumors [19]. The c-Myc protein regulates ribosome biogenesis as well as cooperates with the PI3K/AKT/mTOR pathway to promote the survival of metastatic CRPC cells [17]. Elevated expression of AR splice variant AR-V7 is an important mechanism for developing resistance to enzalutamide and abiraterone [13]. The c-Myc promotes the AR gene transcription and increases the stability of the full-length AR and AR-V7 [20]. Suppression of c-Myc restored sensitivity of enzalutamide-resistant PCa cells to enzalutamide treatment [20]. Pharmaceutical inhibition of c-Myc induces apoptosis in enzalutamide-resistant PCa cells [21, 22]. These observations suggested that c-Myc is essential for enzalutamide-resistant PCa cells. Our study suggested that GRT treatment significantly repressed the protein abundance of c-Myc in enzalutamide-resistant PCa cells. We observed that overexpression of c-Myc enhanced the resistance of C4-2 MDV3100r cells under GRT treatment while knockdown of c-Myc in PC-3 increased the sensitivity of PC-3 cells to GRT treatment. We also observed that level of c-Myc protein expression correlated to resistance of PCa cells to GRT treatment. These findings suggested that c-Myc is one of the main targets of GRT treatment in enzalutamide-resistant PCa cells. The expression level of c-Myc in enzalutamide-resistant prostate tumors may vary in different patients and we predict that those enzalutamide-resistant prostate tumors with lower expression level of c-Myc will be more sensitive to GRT treatment. We also predict that combined treatment of GRT with pharmaceutical inhibitor targeting c-Myc will enhance the regression of tumors.

Additionally, GRT treatment suppressed the expression of proteins involved in cell cycle regulation and cell survival, such as Skp2, cyclin D1, and Cdk1. Cdk1-cyclin complexes are important in the regulation of cell entry and progression to the S and G2/M phases during cell cycle [23]. Skp2 targets $p27^{Kip1}$ by phosphorylating $p27^{Kip1}$ at T187 for ubiquitination and degradation [24]. Skp2 forms a stable complex with the cyclin A-Cdk2 [25]. The reduction of c-Myc, Akt signaling proteins, Skp2, Cdk1, and cyclin D1 can explain the suppression of proliferation and survival of enzalutamide-resistant PCa cells by GRT.

Phosphorylation of AR has been reported to play critical roles in regulating AR function and AR stability. Phosphorylation at Ser81 on AR has been reported to stabilize AR and increase the protein expression of AR, the phosphorylation on Ser81 is regulated by Cdk1 [26] and Cdk5 [27]. For certain CRPC cells, elevation of CDK1 activity is a mechanism to increase AR expression and stability in response to low androgen levels in androgen-deprivation therapy [26]. Mutation of S81A on AR blocks its interaction with CDK5, reduces nuclear localization of AR, destabilizes protein level of AR, and decreases proliferation of PCa cells [27]. In our current study, we observed that GRT treatment suppressed protein expression of AR, phospho-AR (S81), Cdk1, as well as the protein expression of AR's target gene PSA in AR-

positive C4-2 MDV3100r cells. Immunofluorescent microscopy revealed that GRT treatment dose-dependently reduced the abundance of AR proteins both in nucleus and cytoplasm of C4-2 MDV3100r cells. Treatment with cycloheximide confirmed that GRT treatment reduced the stability of AR. The reduction of expression, downstream signaling and stability of AR by GRT will impair the survival and proliferation of C4-2 MDV3100r cells. According to these observations, we predict that enzalutamide-resistant prostate tumors expressing AR will be more sensitive to GRT treatment as compared to those AR-negative enzalutamide-resistant prostate tumors. Interesting, we observed that AR proteins accumulate in the nucleus and PSA expression is relatively high in the enzalutamide-resistant PCa cells even in the absence of androgen. The androgen-independent activation of AR and PSA may be an important mechanism for the proliferation and survival of enzalutamide-resistant PCa cells.

In conclusion, our study suggested that the green rooibos extract, GRT, suppressed the cell proliferation and survival of enzalutamide-resistant PCa cells via inhibition of c-Myc, PI3K-Akt signaling as well as suppressing the expression, stability, and downstream signaling AR. GRT is be a potential adjuvant therapy for PCa patients with enzalutamide-resistance.

## Supporting information

**S1 Fig. HPLC analysis of components in GRT.** The GRT is the same production batch lot used in our previous study [7] and this figure is the same as Fig 1 in [7]. The components of GRT was analyzed at (A) 288 and (B) 350 nm. The content value of each numbered compounds (1, PPAG; 2, isoorientin; 3, orientin; 4, aspalathin; 5, bioquercetin; 6, vitexin; 7, hyperoside; 8, rutin; 9, isovitexin; 10, isoquercitrin; 11, luteoloside; 12, nothofagin) is indicated in brackets after the number on the chromatograms. Component is expressed as g/100 g GRT. (TIF)

**S1 File.**
(DOCX)

**S1 Raw images.**
(PPTX)

**S1 Dataset.**
(DOCX)

## Acknowledgments

The authors thank Prof. Elizabeth Joubert and Prof. Der-San Chuu for the insightful discussion and experimental assistance. We thank Drs. Cheng-Yuan Kao and Ya-Wen Chen for sharing human cell lines. We also thank the support from the Micro-Western Array core facility of NHRI.

## Author Contributions

**Conceptualization:** Christo J. F. Muller, Yung-Hsi Kao, Shu-Pin Huang, Chia-Yang Li, Chih-Pin Chuu.

**Data curation:** Bi-Juan Wang, Shih-Han Huang, Chih-Pin Chuu.

**Formal analysis:** Chih-Pin Chuu.

**Funding acquisition:** Cheng-Li Kao, Christo J. F. Muller, Chien-Chih Yeh, Li-Jane Shih, Chih-Pin Chuu.

**Investigation:** Bi-Juan Wang, Shih-Han Huang, Cheng-Li Kao, Ya-Pei Wang, Hui-Chin Wen.

**Methodology:** Christo J. F. Muller, Kai-Hsiung Chang, Hui-Chin Wen, Chien-Chih Yeh, Li-Jane Shih.

**Project administration:** Chih-Pin Chuu.

**Resources:** Christo J. F. Muller, Kai-Hsiung Chang, Chih-Pin Chuu.

**Supervision:** Chih-Pin Chuu.

**Validation:** Bi-Juan Wang, Shih-Han Huang, Cheng-Li Kao, Chien-Chih Yeh, Li-Jane Shih.

**Visualization:** Christo J. F. Muller, Ya-Pei Wang.

**Writing – original draft:** Chih-Pin Chuu.

**Writing – review & editing:** Christo J. F. Muller, Yung-Hsi Kao, Shu-Pin Huang, Chia-Yang Li, Chih-Pin Chuu.

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
