## [Decision Letter · Decision Letter 0]

12 Apr 2022

PONE-D-22-04289Aspalathus linearis suppresses cell survival and proliferation of enzalutamide-resistant prostate cancer cells via inhibition of c-Myc and stability of androgen receptorPLOS ONE

Dear Dr. Chuu,

Thank you for submitting your manuscript to PLOS ONE. After careful consideration, we feel that it has merit but does not fully meet PLOS ONE’s publication criteria as it currently stands. Therefore, we invite you to submit a revised version of the manuscript that addresses the points raised during the review process. A number of issues were raised by both reviewers which should be addressed if the authors wish to submit a revised manuscript. 

We look forward to receiving your revised manuscript.

Kind regards,

Salvatore V Pizzo

Academic Editor

PLOS ONE

Journal Requirements:

2. We understand that you purchased the extract from a vendor for this study. For purposes of reporting, we request that you provide additional details as to the source of this material (please see http://journals.plos.org/plosone/s/criteria-for-publication#loc-3 for more information).

Reviewers' comments:

Reviewer's Responses to Questions

**Comments to the Author**

1. Is the manuscript technically sound, and do the data support the conclusions?

Reviewer #1: Yes

Reviewer #2: Yes

2. Has the statistical analysis been performed appropriately and rigorously? 

Reviewer #1: Yes

Reviewer #2: Yes

3. Have the authors made all data underlying the findings in their manuscript fully available?

Reviewer #1: Yes

Reviewer #2: Yes

4. Is the manuscript presented in an intelligible fashion and written in standard English?

Reviewer #1: Yes

Reviewer #2: No

5. Review Comments to the Author

Reviewer #1: The manuscript by Huang et describes the growth inhibitory effect of green rooibos extract on the proliferation of the castration resistant C4-2 prostate cancer cells, a derivative which is Enza resistant and the androgen receptor negative expressing PC3 cell line. Authors continue with the androgen receptor positive C4-2 cells and reveal that the extract induces apoptosis. Interestingly, some factors are reduced in protein level including c-Myc. Overexpression of c-Myc reduced extract-mediated growth inhibition.

This is a continuation of experiments of the group previously revealed that the rooibos extract inhibits castration resistance prostate cancer cells through inhibition of YAP and AKT signaling.

Therefore, some additional novelties and specificity must be added to the manuscript

Major points:

1. Authors show reduction of cell viability by the extract in prostate cancer cell lines. This may also indicate an unspecific effect by using 100microgram per ml. Authors should reveal a cell specific response by treatment with a primary explant or primary, non-immortalized, human cells with the same culture condition that is resistant to the treatment.

The general question is whether these high doses will lead to toxicity in non-cancerous cells?

2. Further, the data suggest that those cell lines with c-Myc overexpression are resistant to the treatment. Meaning that those cancer cells, which represent the more aggressive types, which overexpress c-Myc, are treatment resistant. Please discuss this issue.

3. Unclear is why authors focus on androgen receptor degradation, while they reveal that the androgen receptor-negative expressing PC3 cell line is also inhibited in growth by the extract. It would be more helpful to make a general statement to overexpress c-Myc in PC3 cells to reveal whether the rooibos extract targets c-Myc also in these cell lines and whether c-Myc overexpression can ameliorate the apoptotic effect. Thus, reveal in PC3 cells whether rooibos extract targets c-Myc downregulation and by overexpression whether c-Myc is the key factor for rooibos-mediated growth inhibition.

Error bars in Fig. 5B with no GRT are missing.

Label “Actin” with a capital “A”.

Reviewer #2: Enzalutamide is commonly used in treating prostate cancer (PCa) patients, but frequently results in resistance in PCa patients. In this manuscript, the authors examined the effect of a pharmaceutical-grade green rooibos extract (GRT, with 12.8% aspalathin) on the suppression of the survival and proliferation of enzalutamide-resistant prostate cancer (PCa) cells as well as the mechanism. By using two enzalutamide-resistant prostate cancer cell lines C4-2 MDV 3100r and PC-3, they demonstrated that both are resistant to 10uM MDV compared to the C4-2 cells, and identified the IC50 concentrations of GRT in C4-2 MDV 3100r and PC-3 both cell lines. Furthermore, GRT treatment in a dose-dependent manner can reduce cell proliferation, decrease colony formation in soft agar and induce apoptosis by comet assay in C4-2 MDV 3100r cells. In the mechanism study by examining a panel of signaling proteins in cell cycle, apoptosis, cell survival, and androgen receptor (AR) signaling in C4-2 MDV 3100r treated with different concentrations of GRT, the authors found the increasing dose of GRT reduces the levels of the proteins in cell cycle and cell survival as well as increases the proteins in apoptosis. In addition, they indicated that Myc overexpression elevates the resistance in C4-2 MDV 3100r cells under GRT treatments. Moreover, they showed that GRT treatment diminish AR expression, stability, and PSA expression in C4-2 MDV 3100r cells. The authors suggest that GRT may serve as a potential adjuvant therapeutic agent for enzalutamide-resistant prostate cancer. Overall, this is a nice and thorough work providing the mechanism study of GRT on enzalutamide-resistant prostate cancer cells.

However, there are some typos or grammatical errors as below need to be corrected.

1. In line 56, 280, and 347, it should be “Interestingly” not “Interesting”.

2. In line 78, delete “in” before every year.

3. Line 82, delete “will”.

4. Line 85, should be “patients”.

5. Line 108, play “an” essential role.

6. All the antibodies and special reagents or kits should be included with the catalogue number in the Materials and Methods.

7. The first paragraph in the first section of the results from line 208 to 217 can be moved to the Materials and Methods.

8. The more correct way of labeling should be “phospho-PDK1 (S241)” in line 253 and 308, and 427, similarly like “phosphor-AR (S81)” in line 254 and 428.

9. Line 276, “reduces”.

10. Line 292, “mediated”.

11. Line 432, “enhanced”.

In addition, the images of the figure 1D, 1E, 2C, and 6A should include the scale bar.

6. PLOS authors have the option to publish the peer review history of their article (what does this mean?). If published, this will include your full peer review and any attached files.

Reviewer #1: No

Reviewer #2: No

---

## [Author Response · Author response to Decision Letter 0]

27 May 2022

05/26/2022

Dear Editor,

 We are here re-submitting the paper “Aspalathus linearis suppresses survival and proliferation of enzalutamide-resistant prostate cancer cells via inhibition of c-Myc and stability of androgen receptor” by Bi-Juan Wang and Shih-Han Huang et al. for consideration for publication as research paper. No related content of this research has been submitted or published elsewhere. We have revised the manuscript and added two experiments according to the suggestions made by the reviewer, as well as answered all the questions raised by the reviewers. We also included our original Western blot scans as well as provide a file of minimal data set as supplemental material for review purpose. We have revised the manuscript to meet PLOS ONE's style requirements. The pharmaceutical-grade green rooibos extract (GRT, containing 12.78% aspalathin) being used in this study was extracted and produce by the co-author Prof. Christo J.F. Muller from Medical Research Council of South Africa. We included the HPLC analysis of this batch of GRT as supplemental figure. We have corrected the funding information in the submission system so that the information is now consistent with that in the manuscript. We hope that the current revised manuscript fulfill the requirement of PLOS One.

The following is a response to the reviewers’ concerns and questions.

Reviewer #1: The manuscript by Huang et describes the growth inhibitory effect of green rooibos extract on the proliferation of the castration resistant C4-2 prostate cancer cells, a derivative which is Enza resistant and the androgen receptor negative expressing PC3 cell line. Authors continue with the androgen receptor positive C4-2 cells and reveal that the extract induces apoptosis. Interestingly, some factors are reduced in protein level including c-Myc. Overexpression of c-Myc reduced extract-mediated growth inhibition. This is a continuation of experiments of the group previously revealed that the rooibos extract inhibits castration resistance prostate cancer cells through inhibition of YAP and AKT signaling.

Therefore, some additional novelties and specificity must be added to the manuscript

Answer: We thank reviewer for the affirmative comment.

Major points:

1. Authors show reduction of cell viability by the extract in prostate cancer cell lines. This may also indicate an unspecific effect by using 100microgram per ml. Authors should reveal a cell specific response by treatment with a primary explant or primary, non-immortalized, human cells with the same culture condition that is resistant to the treatment. The general question is whether these high doses will lead to toxicity in non-cancerous cells?

Answer: We thank the reviewer for raising this important question. We now included the data of HUVEC (human umbilical vein endothelial cells) and HEK293 (human embryonic kidney cells 293). The HUVEC cells are normal human primary cells and HEK293 cells is an established cell line with mutant version of SV40 large T antigen. Proliferation of HUVEC cells was not affected by 10-75 μg/ml GRT and was relatively resistant to 100 μg/ml GRT, while HEK293 was not affected by GRT at all. This observation suggested that enzalutamide-resistant PCa cells were more sensitive to GRT treatment than non-cancerous human cell lines.

2. Further, the data suggest that those cell lines with c-Myc overexpression are resistant to the treatment. Meaning that those cancer cells, which represent the more aggressive types, which overexpress c-Myc, are treatment resistant. Please discuss this issue.

Answer: We thank the reviewer for raising this very interesting question. We performed a new Western blotting assay to compare the protein expression of c-Myc in C4-2 MDV3100r and PC-3 cells being treated with 0 or 100 μg/ml GRT. PC-3, which was more resistant to GRT treatment, expressed higher level of c-Myc protein as compared to C4-2 MDV3100r (new Fig. 6C). We also added a new c-Myc knockdown experiment on PC-3 cells to examine if knockdown of c-Myc increases the sensitivity of PC-3 cells to GRT treatment. Knockdown of c-Myc with siRNA in PC-3 cells increased the sensitivity of PC-3 cells to GRT treatment (new Fig. 6A, 6B). We added the following discussion in the Discussion, “Our study suggested that GRT treatment significantly repressed the protein abundance of c-Myc in enzalutamide-resistant PCa cells. We observed that overexpression of c-Myc enhanced the resistance of C4-2 MDV3100r cells under GRT treatment while knockdown of c-Myc in PC-3 increased the sensitivity of PC-3 cells to GRT treatment. We also observed that level of c-Myc protein expression correlated to resistance of PCa cells to GRT treatment. These findings suggested that c-Myc is one of the main targets of GRT treatment in enzalutamide-resistant PCa cells. The expression level of c-Myc in enzalutamide-resistant prostate tumors may vary in different patients and we predict that those enzalutamide-resistant prostate tumors with lower expression level of c-Myc will be more sensitive to GRT treatment. We also predict that combined treatment of GRT with pharmaceutical inhibitor targeting c-Myc will enhance the regression of tumors.”

3. Unclear is why authors focus on androgen receptor degradation, while they reveal that the androgen receptor-negative expressing PC3 cell line is also inhibited in growth by the extract. It would be more helpful to make a general statement to overexpress c-Myc in PC3 cells to reveal whether the rooibos extract targets c-Myc also in these cell lines and whether c-Myc overexpression can ameliorate the apoptotic effect. Thus, reveal in PC3 cells whether rooibos extract targets c-Myc downregulation and by overexpression whether c-Myc is the key factor for rooibos-mediated growth inhibition.

Answer: We thank the reviewer for raising this important question. A majority portion of CRPC tumors still express AR. We therefore examine if GRT may suppress the protein expression, phosphorylation, signaling transduction, and stability of AR in the AR-positive C4-2 MDV3100r cells. However, as the reviewer pointed out, PC-3 cells are AR-negative PCa cells. As PC-3 cells express much higher c-Myc protein as compared to C4-2 MDV3100r cells (new Fig. 6C), we chose to knock down c-Myc in PC-3 cells and to overexpress c-Myc in C4-2 MDV3100r cells to determine if c-Myc is one of the main targets of GRT in CRPC cells. We observed that overexpression of c-Myc enhanced the resistance of C4-2 MDV3100r cells under GRT treatment while knockdown of c-Myc in PC-3 increased the sensitivity of PC-3 cells to GRT treatment (new Fig. 6A, 6B). These observations suggested that c-Myc is one of the main targets of GRT in enzalutamide-resistant PCa cells.

Error bars in Fig. 5B with no GRT are missing.

Answer: We thank the reviewer for noticing this mistake. We now added the error bars into the condition with GRT treatment in Fig. 5B.

Label “Actin” with a capital “A”.

Answer: We thank the reviewer for noticing this mistake. We now corrected all “actin” to “Actin”.

Reviewer #2: Enzalutamide is commonly used in treating prostate cancer (PCa) patients, but frequently results in resistance in PCa patients. In this manuscript, the authors examined the effect of a pharmaceutical-grade green rooibos extract (GRT, with 12.8% aspalathin) on the suppression of the survival and proliferation of enzalutamide-resistant prostate cancer (PCa) cells as well as the mechanism. By using two enzalutamide-resistant prostate cancer cell lines C4-2 MDV 3100r and PC-3, they demonstrated that both are resistant to 10uM MDV compared to the C4-2 cells, and identified the IC50 concentrations of GRT in C4-2 MDV 3100r and PC-3 both cell lines. Furthermore, GRT treatment in a dose-dependent manner can reduce cell proliferation, decrease colony formation in soft agar and induce apoptosis by comet assay in C4-2 MDV 3100r cells. In the mechanism study by examining a panel of signaling proteins in cell cycle, apoptosis, cell survival, and androgen receptor (AR) signaling in C4-2 MDV 3100r treated with different concentrations of GRT, the authors found the increasing dose of GRT reduces the levels of the proteins in cell cycle and cell survival as well as increases the proteins in apoptosis. In addition, they indicated that Myc overexpression elevates the resistance in C4-2 MDV 3100r cells under GRT treatments. Moreover, they showed that GRT treatment diminish AR expression, stability, and PSA expression in C4-2 MDV 3100r cells. The authors suggest that GRT may serve as a potential adjuvant therapeutic agent for enzalutamide-resistant prostate cancer. Overall, this is a nice and thorough work providing the mechanism study of GRT on enzalutamide-resistant prostate cancer cells.

Answer: We thank reviewer for the affirmative comment.

However, there are some typos or grammatical errors as below need to be corrected.

1. In line 56, 280, and 347, it should be “Interestingly” not “Interesting”.

2. In line 78, delete “in” before every year.

3. Line 82, delete “will”.

4. Line 85, should be “patients”.

5. Line 108, play “an” essential role.

6. All the antibodies and special reagents or kits should be included with the catalogue number in the Materials and Methods.

7. The first paragraph in the first section of the results from line 208 to 217 can be moved to the Materials and Methods.

8. The more correct way of labeling should be “phospho-PDK1 (S241)” in line 253 and 308, and 427, similarly like “phosphor-AR (S81)” in line 254 and 428.

9. Line 276, “reduces”.

10. Line 292, “mediated”.

11. Line 432, “enhanced”.

Answer: We sincerely apologize for these typos in the manuscript and we appreciate the reviewer for pointing out these typos. We have now corrected all these typos.

In addition, the images of the figure 1D, 1E, 2C, and 6A should include the scale bar.

Answer: We thank the reviewer for finding these mistakes. We have now added scale bars into these figures.

Sincerely,

Chih-Pin Chuu, Ph.D., Associate Investigator/Associate Professor

Institute of Cellular and System Medicine, National Health Research Institutes, Taiwan

---

## [Decision Letter · Decision Letter 1]

20 Jun 2022

Aspalathus linearis suppresses cell survival and proliferation of enzalutamide-resistant prostate cancer cells via inhibition of c-Myc and stability of androgen receptor

PONE-D-22-04289R1

Dear Dr. Chuu,

We’re pleased to inform you that your manuscript has been judged scientifically suitable for publication and will be formally accepted for publication once it meets all outstanding technical requirements.

Kind regards,

Salvatore V Pizzo

Academic Editor

PLOS ONE

Additional Editor Comments (optional):

Reviewers' comments:

Reviewer's Responses to Questions

**Comments to the Author**

1. If the authors have adequately addressed your comments raised in a previous round of review and you feel that this manuscript is now acceptable for publication, you may indicate that here to bypass the “Comments to the Author” section, enter your conflict of interest statement in the “Confidential to Editor” section, and submit your "Accept" recommendation.

Reviewer #1: All comments have been addressed

Reviewer #2: All comments have been addressed

2. Is the manuscript technically sound, and do the data support the conclusions?

Reviewer #1: Yes

Reviewer #2: Yes

3. Has the statistical analysis been performed appropriately and rigorously? 

Reviewer #1: Yes

Reviewer #2: Yes

4. Have the authors made all data underlying the findings in their manuscript fully available?

Reviewer #1: Yes

Reviewer #2: Yes

5. Is the manuscript presented in an intelligible fashion and written in standard English?

Reviewer #1: Yes

Reviewer #2: Yes

6. Review Comments to the Author

Reviewer #1: Authors addressed all crtical points in a satisfactory manner.

Authors addressed all crtical points in a satisfactory manner

Reviewer #2: The author has provided the additional data for the questions brought by the reviewers and corrected all the typos and grammatical errors, as well as included the scale bars. The manuscript has met the requirement now.

7. PLOS authors have the option to publish the peer review history of their article (what does this mean?). If published, this will include your full peer review and any attached files.

Reviewer #1: No

Reviewer #2: No

---

## [Editor Report · Acceptance letter]

23 Jun 2022

PONE-D-22-04289R1 

*Aspalathus linearis* suppresses cell survival and proliferation of enzalutamide-resistant prostate cancer cells via inhibition of c-Myc and stability of androgen receptor 

Dear Dr. Chuu:

I'm pleased to inform you that your manuscript has been deemed suitable for publication in PLOS ONE. Congratulations! Your manuscript is now with our production department. 

Kind regards, 

on behalf of

Dr. Salvatore V Pizzo 

Academic Editor

PLOS ONE